# Adoption of Food Species Mixtures from Farmers' Perspectives in Germany: Managing Complexity and Harnessing Advantages

Johannes Timaeus [1,*], Ties Ruigrok [2], Torsten Siegmeier [3] and Maria Renate Finckh [1]

1 Department of Ecological Plant Protection, University of Kassel, Nordbahnhofstr. 1 a, 37213 Witzenhausen, Germany; mfinckh@uni-kassel.de
2 Aeres Group, Wisentweg 10, 8251 PC Dronten, The Netherlands; t.ruigrok@aeres.nl
3 Department of Farm Management, University of Kassel, Steinstr. 1 9, 37213 Witzenhausen, Germany; siegmeier@uni-kassel.de
* Correspondence: johannes.timaeus@uni-kassel.de

**Abstract:** Many agronomic studies have shown the advantages of species mixtures (SM), but for food grain production, they represent only a small niche. Empirical studies that investigate reasons for SM adoption in food grain production are scarce. Here we present an in-depth study based on qualitative expert interviews with nine farmers. By means of interpretative analysis and reconstruction, socially shared models of SM adoption were built to identify the five main factors for SM adoption: (1) perceived relative mixture performance compared to sole crops, (2) suitability within the farm context (3), challenges and opportunities in mixture management due to increased complexity, (4) knowledge and technology as resources to handle mixture management and (5) quality standards in the food value chain. Relative performance was perceived as higher for SM than for sole crops for crop protection, nutrient efficiency, farm diversification, total yield stability and grain quality. The yield stability of individual crop species in SM was perceived as lower and grain impurities higher, requiring increased separation efforts. The economic potential of SM was perceived as highly variable, depending on crop value and post-harvest efforts to attain food quality. Reconstructing the mixture management process revealed that the interspecific plant interactions and emergent mixture attributes increased the cropping system complexity and affected the entire farming process. Adopting SM required knowledge about species interactions, mixture attributes and equipment settings. Large knowledge gaps for food SM were identified. The complexity of SM also provided opportunities for farmers to design mixtures that allow competition control (alternate rows) or avoid separation (relay mixtures). The main conclusions are: (1) increased complexity is a basic property of SM compared to sole crops, enabling advantages and increasing the option space to develop new sustainable cropping systems, (2) specific knowledge and technology are required for SM and are not accessible for most farmers, requiring new information channels and (3) new food SM should be developed more systematically, taking into account mixture properties and their effects on the farming process, as well as needs from the food value chain.

**Keywords:** diversified farming; intercrops; value chain; food system; agroecology

## 1. Introduction

Species mixtures (SM) or intercrops are a key diversification strategy in agriculture to reduce external inputs for fertilization and crop protection [1,2]. Legume-cereal SM in temperate agroecosystems increases the use of soil-derived and biologically fixed nitrogen, reducing the need for energy-intensive synthetic fertilizer production [3]. Yields of legume-cereal SMs can be more stable than yields of the respective sole crops [4] and considerable yield gains of mixtures were found in low and high-input farming systems [5]. In a recent experimental study, wheat-pea mixtures outperformed the respective sole crops within a multifunctional evaluation framework, including yields, resource efficiency, crop

quality and crop protection, especially under challenging environmental conditions [6]. Despite this potential, in industrialized agriculture, SM are mostly confined to the niches of grassland and cover crop mixtures [7]. SM for food grains are a small niche even in organic agriculture. Lentil-cereal mixtures represent a noticeable exception, being one of the few food grain SMs used in agricultural practice in Europe [8]. This raises the question of what is hindering the wider adoption of species mixtures in food cropping and how these barriers can be overcome in a European farming context.

On the one hand, economic and public policy aspects [9], and, on the other hand, plant traits relevant for breeding SM in a food supply chain context have been identified, for example, the susceptibility of pea cultivars for grain splitting, enforcing increased separation efforts [10]. Bybee-Finley and Ryan review SM advances in a European industrial agricultural context but focus on experimental methodology [7]. Hauggaard-Nielsen et al. [11] explore experiences and opportunities of participatory multi-actor approaches to foster SM adoption. However, social scientific studies that investigate the process of adoption of SM for food grain production from a farmers' perspective are missing.

In Germany, Lemken et al. [12] conducted a quantitative survey to investigate the underlying causes impeding species mixture implementation. According to this study, psychosocial factors, such as the perception of technical barriers and advantages of the mixtures are important. Due to the quantitative nature and the very low adoption rate of species mixtures in Germany, their study neither provides deeper insights into the specific causes and processes influencing the diffusion of SM in practice nor about the levers fostering SM implementation and innovation. The processes leading to the adoption of innovations are intricate and conditioned by the subjective realities of farmers, as well as the broader socio-economic conditions [13]. Explorative and interpretative approaches in social sciences are able to reconstruct social processes in detail [14,15] and identify levers to foster mixture adoption. Thus far, we are not aware of any interpretative, in-depth reconstruction of the adoption of SM for food grains in Germany.

Three main research questions were addressed in this study: (1) What are the underlying factors influencing mixture adoption, and how do they interact from a farmer's perspective? (2) How do SM influence the farming process from sowing to post-harvest procedures? (3) Based on the main factors for SM adoption and the farming process, what are the main levers to foster adoption and innovation?

Following Corbin and Strauss [14], an empirically grounded approach allowed us to reconstruct the processes of mixture implementation in-depth and to construct qualitative socially shared models of these processes [16], enabling a deeper understanding of mixture adoption in complex contexts. The research presented here was part of the EU project ReMIX aimed at integrating multiple actors in research and practice into the research process [11,17].

## 2. Research Process, Materials and Methods

### 2.1. Participatory Process

The participatory multi-actor process created access to experience-based knowledge and social communication networks. Our first step was a workshop with farmers, advisors and technicians to discuss the main challenges and opportunities for SM in Germany to establish first contacts for on-farm experiments and for interviews with farmers experienced in SM. On-farm experiments with species mixtures were conducted on four farms, establishing a stronger link to farmers and resulting in fruitful informal discussions about challenges and strategies for mixtures in practice. Here we tested wheat-pea species mixtures that are promising in terms of yield gains and increased protein contents of wheat grains. This allowed us to enter the social research field of practical mixture farming and adoption and learn, understand and analytically reconstruct the processes from the farmer's perspectives and gain openness and trust for exchange [14].

*2.2. Interview Guide Development and Data Gathering*

　　Formal narratives were written each season, documenting the multi-actor process. Further empirical data were gathered by semi-structured expert interviews [18]. Expert interviews draw on the experience-based knowledge of interviewees to uncover mechanisms of social processes [19], here the diffusion of SM in practice. In these interviews, cereal-grain mixtures for food were focused on.

　　The interview guide addressed the following main themes: (1) opening and motivation for mixture adoption, (2) characteristics of SM compared to sole crop, (3) sources of information used, (4) special cultivation and processing aspects of mixed cropping cultivation, (5) use of SM, quality aspects and economic efficiency and (6) conclusion of the interview. The full interview guide can be found in the Supplementary Materials. The interview topics were formulated as open questions to allow the experts to provide their knowledge and experience. If required, more focused questions were posed to more deeply explore the detailed expert knowledge and perspective [18]. A pilot version of the interview guide was developed within the research team, it was tested and further refined to improve its reliability as a research tool, balancing the focus on the addressed research questions while maintaining openness for the experiential knowledge and perspectives of the farmers. The prerequisite for a farmer to be an expert and to qualify as an interviewee was experience with food grain mixtures for at least three years. All farmers designed and mixed their food SM on their own since they are not commercially available as ready-made mixtures.

　　The interviews were conducted in 2019 and 2020. As farmers with experience in food grain mixture cultivation are rare and difficult to find, they were identified following the snowball principle [20]. In total, nine farmers were interviewed, wight organic and one conventional (Table 1). Organic farmers farmed according to EU organic standards set by Regulation (EC) No 834/2007. While the number of farmers sampled is low compared to quantitative methods, the interview analysis showed considerable theoretical saturation as a key quality standard in qualitative research [14]. This was indicated by decreasing new insights and the repetition of factors for SM adoption during the last interviews conducted. An overview of the cultivated mixtures and seed traits is provided in Figure 1.

**Table 1.** Overview of the farms of the interviewees. F: farmer, S: federal state, FA: farming area, FS: farming system (O: organic, C: conventional), SQ: soil quality in bp based on the German soil classification system ranging from 1 to 100 bp, ME: mixture experience, SCT: specific cultivation technology refers to cultivation equipment specifically designed for species mixtures, SHT: specific post-harvest technology refers to post-harvest equipment specifically designed for species mixtures, B: Bavaria, BW: Baden-Württemberg, H: Hessen, LS: Lower Saxony, NRW: North Rhine-Westphalia, NA: missing information.

| F | S | FA (ha) | FS | SQ (bp) | ME (Years) | Species Mixture Used | SCT | SHT | Customers |
|---|---|---|---|---|---|---|---|---|---|
| 1 | LS | 140 | O | 49 | 28 | winter rye-vetch, lentil-barley, linseed-barley, oat-pea, barley pea | none | Pre-cleaning, drying, fine cleaning, optical separation technology, weight selector, contract processing for food purposes | Natural food wholesale, mills, oil mills, seed distribution, livestock farmers |
| 2 | H | 67 | O | 30–65 | 31 | linseed-false flax | none | Pre-cleaning and drying technology, fine screen cleaning technology | Bakers, oil mills, seed companies |
| 3 | NRW | 110 | O | 11–42 | 48 | canola-vetch | modified seeder, cage roller, roller hoe | Pre-cleaning, mobile drying, fine cleaning | Oil mills |
| 4 | BW | 20 | C | 27–32 | 7 | lentil-false flax | none | Done by contractor | Food retailer |

**Table 1.** *Cont.*

| F | S | FA (ha) | FS | SQ (bp) | ME (Years) | Species Mixture Used | SCT | SHT | Customers |
|---|---|---|---|---|---|---|---|---|---|
| 5 | BW | 350 | O | 40–70 | 8 | winter wheat-pea | none | Pre-cleaning, drying, screen cleaning | Producers' association |
| 6 | BW | 82 | O | 25–50 | 15 | lentil-barley | none | Drying and separation technology, contract drying and separation | Producer group, brewery |
| 7 | B | 20 | O | 30–40 | 14 | false flax-oat | none | Drying and fine sieves/screens | Oil mill selling to regional market |
| 8 | B | 60 | O | 60 | 3 | lupin-barley | none | Pre-cleaning, car drying | Producer association |
| 9 | BW | 95 | O | 15–45 | 9 | lentil-oat | none | Pre-cleaning, drying, fine cleaning: weight and optical separation technology, | NA |

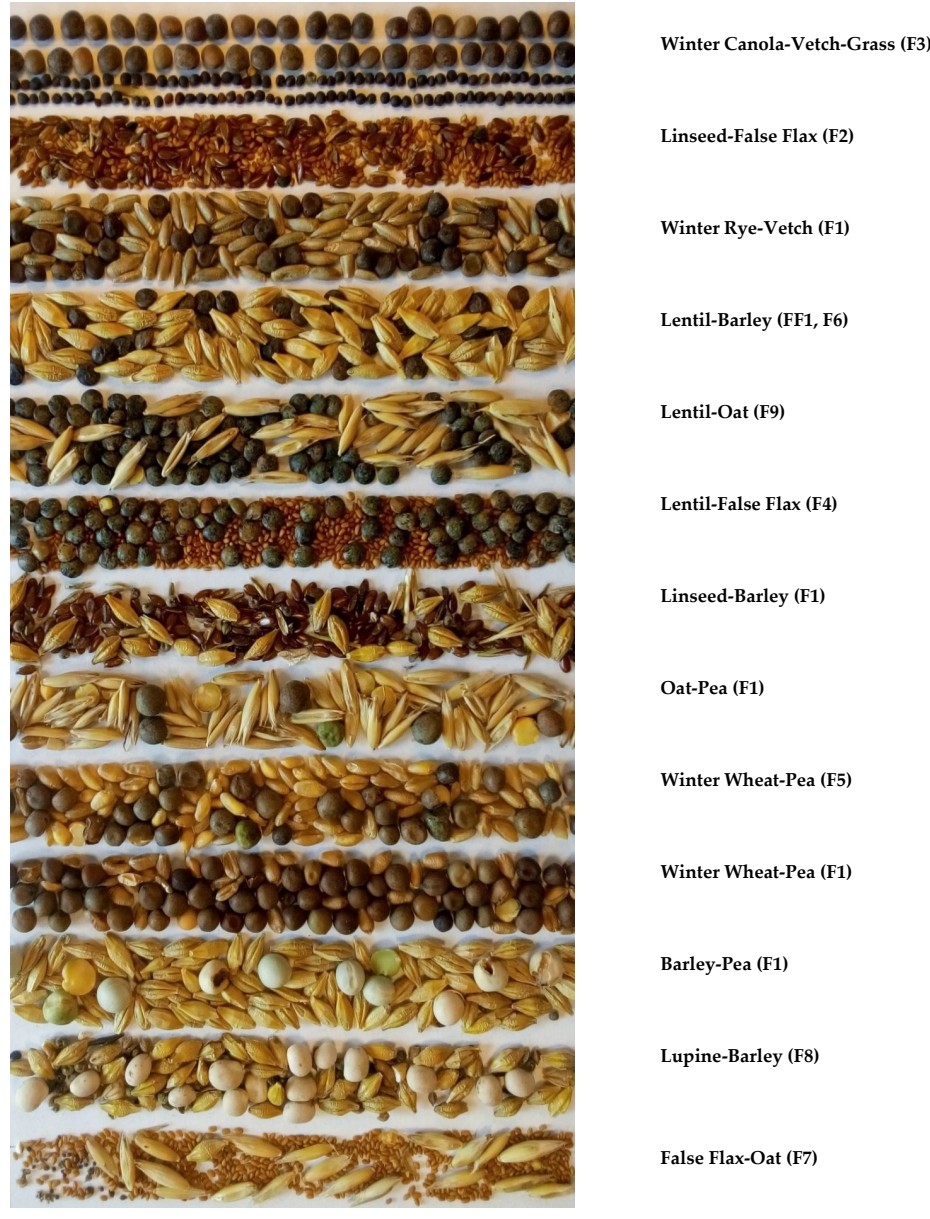

Winter Canola-Vetch-Grass (F3)

Linseed-False Flax (F2)

Winter Rye-Vetch (F1)

Lentil-Barley (FF1, F6)

Lentil-Oat (F9)

Lentil-False Flax (F4)

Linseed-Barley (F1)

Oat-Pea (F1)

Winter Wheat-Pea (F5)

Winter Wheat-Pea (F1)

Barley-Pea (F1)

Lupine-Barley (F8)

False Flax-Oat (F7)

**Figure 1.** Mixtures sampled from the interviewed farmers. Photo by Ties Ruigrok.

*2.3. Interview Analysis and Reconstruction of Socially Shared Representations*

Interview data were recorded, transcribed verbatim and analyzed in MAXQDA with qualitative coding methods [18] to identify the factors for the diffusion of SM and the impact of SM on the farming process. In the iterative coding process, new codes were developed deductively to better fit the specific context and reality of the farmers and to develop empirically grounded categories and concepts [14]. The codes were developed in a collaborative process of constant feedback in the research group to enable its inter-subjectivity. For factors of high relevance and where the empirical material was rich and "thick" [21] we analytically zoom in on detail, e.g., for mixture attributes, while for some factors that were identified as crucial, but the material was limited, we remain on a more birds-eye perspective. The main findings were visualized in diagrams constituting social representations, i.e., socially shared models [16] of the mixture adoption process that were reconstructed from the interviews. The empirical material (narratives, interviews, notes on on-farm experiments and conversations) from multiple resources allowed us to investigate the adoption phenomenon from multiple perspectives and to implement a triangulation approach suited to study complex socio-economic phenomena [22].

## 3. Results

From farmers' perspectives, SM adoption was influenced by four main groups of factors and their interactions at the farm level: (1) perceived relative performance (PRP) of mixtures compared to sole crops, (2) suitability of mixtures for the specific farming system, (3) management challenges from mixture design to post-harvest procedures and (4) knowledge and technology as resources deployed for mixture management at the farm level. While the farm level was the focus of the study, clearly, interactions with actors in the value chain for food and the contextual food system had strong influences on mixture adoption on-farm, representing the fifth main factor for mixture adoption (Figure 2).

*3.1. Perceived Relative Performance of Mixtures*

Compared to sole crops, the PRP of SM was a key motivation for the integration of mixtures into the farming system. PRP was perceived higher for SM than sole crops for (1) improved nutrient efficiency and acquisition through biological fixation in the case of legumes, (2) crop protection and (3) overall higher system diversity. PRP for mixtures was assessed much more critically and mixed for (4) food grain quality, yield stability and (5) economic potential. Frequently, a mixed crop was integrated into the farm because of negative experiences with or assessments of a sole crop (F1, F3, F2, F6, F7, F8). PRP interacts with the suitability of mixtures to the farming system as only under specific conditions was PRP considered to be in favor of mixtures.

3.1.1. Nutrient Efficiency and N-Fixation

In the context of very nutrient-demanding crops, legume mixtures were considered a good strategy for improved nitrogen supply. Frost-sensitive vetch was used by F3 in a SM with winter oilseed canola, whose high nitrogen requirements are otherwise difficult to supply in organic farming. The organic cultivation of canola as a sole crop was considered a challenge, especially with respect to crop health that could be strengthened in mixtures (see below). The incorporation of legumes into a cereal crop to increase nitrogen availability was similarly seen as a key advantage (F1, F3, F5, F6, F9). According to farmers, nitrogen fixation by legumes is enhanced because the non-legumes compete for mineral nitrogen from the soil pushing the legume to rely more on biological nitrogen fixation. Compared to cereal sole crops, the pre-crop effect of SM with legumes was perceived as advantageous (F9, F8). Furthermore, the possibility of using milled unseparated mixtures of oats and pea as forage for livestock-keeping farms was mentioned as a possible fallback strategy to use the mixtures if food-grade quality could not be achieved (F2).

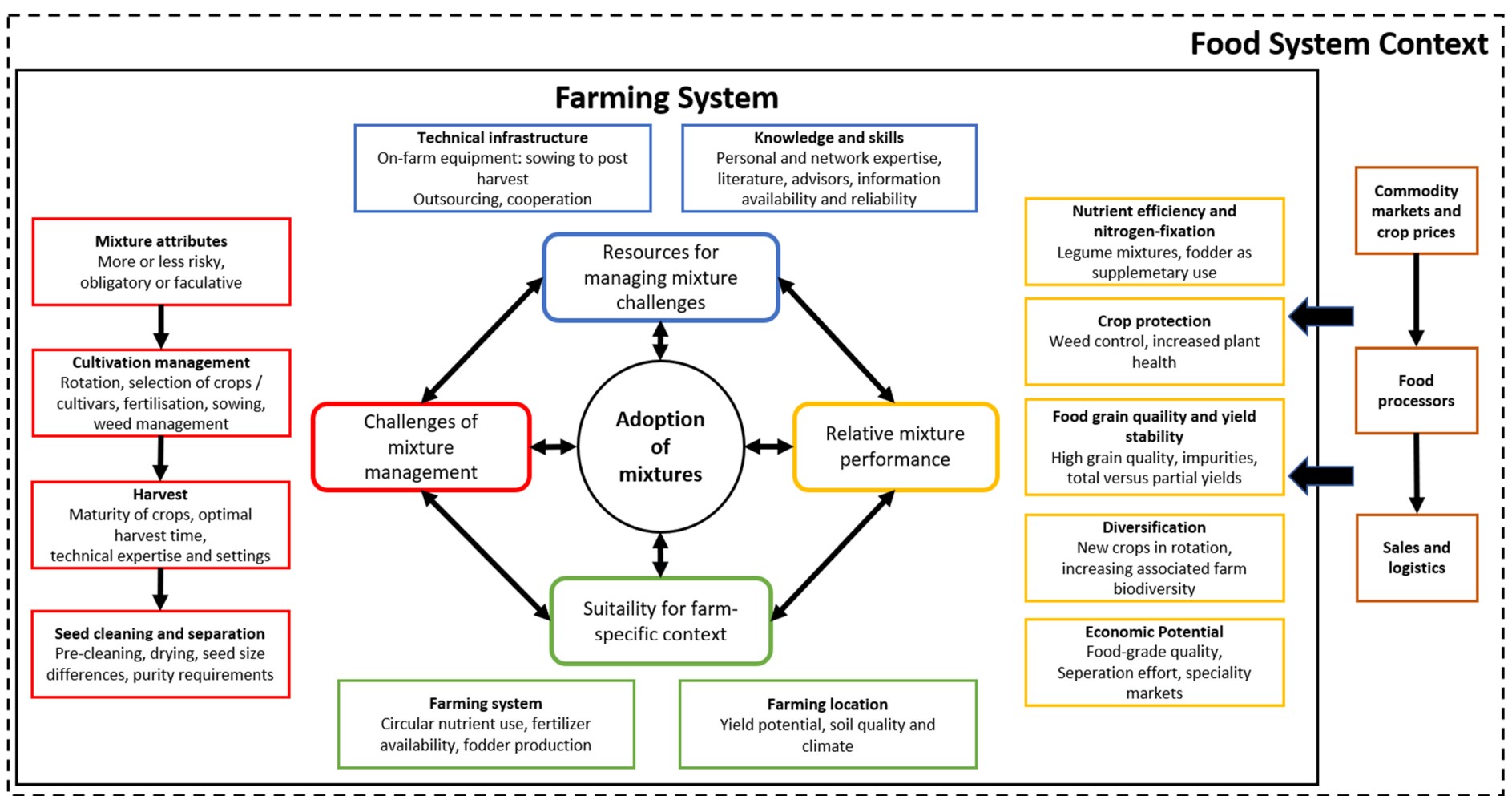

**Figure 2.** Social representation of the main factors and their interactions influencing species mixture adoption for food grain production within a broader food system context. The social representation was interpretatively reconstructed from the interviews with farmers.

### 3.1.2. Crop Protection

Farmers considered reduced weed pressure and simplified weed control a relative advantage of mixtures (F1, F2, F3, F5, F7, F8). "the false flax with its initial rosette can cover much more ground and thereby prevents seed weeds from germinating." (F2). Predominantly, the good health of plants and low pathogen pressure in mixtures were mentioned (F1, F2, F3, F4, F7, F8, F9). For F9, fungal diseases had never "been an issue", or they were considered "minor" in organic farming by F1. Farmers had cognitive models for underlying mechanisms to support their assessment of relative performance advantages of mixtures, including, e.g., distractive effects of the mixed crop partner on pests of the main crop (F2), a higher attraction of pest insects to the mixing partner vetch in canola mixtures (F3) and reduced spreading of diseases in mixtures (F8).

### 3.1.3. Food Grain Quality and Yield Stability

The quality of grains for food was judged to be better in mixtures due to reduced lodging of legumes and/or improved nitrogen supply to the cereals in mixtures. Due to the supporting effect of the mixtures, less soil, dust and sand were visible on the grains in lentil mixtures (F1, F4). The requirements for legumes in the organic food market (F1) or for malting barley (F6) were met with SM after separation. However, attaining food grain quality from mixtures was regarded as a considerable challenge in many cases. F9 reported that due to low hectoliter weight, the food-grade requirements for hulled oats from lentil mixtures were not fulfilled. Quality standards are very high (free of foreign odor or bitter substances) for cold-pressed rapeseed oil, and contamination with weed seeds or mixture partners, such as vetch surviving the frost, are not tolerated by the oil mill as this can influence the sensory characteristics (F3).

Wheat from wheat-pea mixtures did not always meet the demanding quality criteria for baking with respect to remaining impurities after separation (F1). A strategy to meet this challenge was to pre-separate mixtures on-farm while food processors, such as mills, conducted the food-grade separation (F5, F7). The effort required for good separation and purity was estimated to be higher for mixtures than for sole crops. Here, the effort was due to multiple passes of the batches in the cleaning machines (F2, F5, F6, F7, F9). In the case of poor separability, a "disproportionate effort" was to be made with weight and color separators (F9), which was associated with higher costs (F1). In this case, batches had to be sold as animal fodder resulting in critical economic loss (F1).

A higher PRP of mixtures in terms of stability of total yield resulted in a reduced production risk (F1, F2, F3, F5, F7, F8). According to F6: "An advantage is mainly the risk balance. In the species mixture . . . the whole crop stands on two legs, not just one. That means that if something happens to the crop [ . . . ] the other partner can thrive all the better. That is simply safer ". Due to current weather extremes, the benefits of mixtures were considered evident and to be "exciting" (F5). Self-critically, the lack of long-term experience with peas as a sole crop and the associated yield fluctuations were mentioned by one farmer (F7). In contrast, partial yields of the respective mixture partners were reported to be highly variable (see below Section 3.1.5).

### 3.1.4. Diversification of Crops and Associated Organisms

Mixtures enrich the cropping system by extending the rotation with additional crops that were not or hardly ever grown as a sole crop (F1, F3, F7, F9). This includes lentils and peas of the indeterminate growth type, both highly vulnerable to lodging and weeds. F1 explained about mixtures that "it gives you the opportunity to cultivate crops that could not be cultivated at all. So a lentil alone does not work. A long-stemmed winter or spring pea doesn't work either, because the stuff lies flat on the ground." (F1). In this case, the PRP of the mixture is infinitely better than the sole crop. Lupin that was cultivated by F8 in a mixture with barley was also a challenge as a sole crop as well as canola sole crops. In organic agriculture, canola-vetch mixtures facilitated the inclusion of canola into the rotation (F3).

Another perceived relative advantage of mixtures was the promotion of habitats for biodiversity associated with diversified cropping systems. Worth mentioning are rare insect and bird species, as well as the promotion of soil life (F5). This benefit was considered an "additional yield" (F5). Explicitly, the flowering aspect of the mixed-species was mentioned (F5). Phacelia for bees in barley had no negative consequences for separation due to large seed size differences according to F1. In mixed crops, more room for weeds was perceived (F6). "Lentil fields are such ecological highlights, even on our farm, and on our lentil fields there are specimens from the red list. And especially on the lentil fields the biodiversity is huge. In our case there were between 60 and 80 different plant species on one lentil field" (F6).

### 3.1.5. Economic Potential

Altogether, the perception of economic potentials varied with some opportunities being mentioned but also higher costs compared to sole crops for seeds, cleaning and separation, and, in part, additional efforts required to find customers in the value chain for the product. For F1, the economic efficiency of wheat-pea mixtures was perceived better than the wheat sole crop. While the wheat yield was similar to the sole crop, "a few decitons of peas" are harvested in addition, leading to greater profitability (F5). The mixed crop cultivation of lentil and linseed was rated as "much better" than the sole crop (F4). Barley-lentil was described as "[ . . . ] the crop with the highest contribution margin [ . . . ] and with low input." (F6). In contrast, the highest cultivation risk was attributed to lentil mixtures by F1, F4 and F9 due to weather-related fungal diseases and high yield fluctuations observed over several seasons. Lentil mixtures are counted among the expensive crops due to high seed costs (F1). Here, the separation effort must also be put in relation to the total revenue (F1). The cleaning of small quantities was "more labor-intensive" (F1). Compared to a lupine sole crop, the lupine mixture with barley resulted in higher seed costs and lower yields for F8, leading to lower economic efficiency. The reason is that uniform maturing varieties suitable for mixing are lower yielding than unevenly maturing lupin varieties that are also more competitive. Thus, there, the sole crop is more profitable (F8).

A failure to reach food grade quality is not unlikely and reduces mixture profitability. In principle, the value chain was seen as an important factor for the profitability of mixtures by all farmers (F1–F9). This was especially recommended for newcomers. A purchase guarantee (F2) with a fixed price for lentil contract cultivation resulted in a high attractiveness of the cultivation (F6). For further assurance of purchases, active customer loyalty was cultivated. This was done, e.g., by inviting buyers to field visits (F2). The exchange of expertise with the processing baker or the oil mill and personal relationships were found to be important (F2). One of the farmers interviewed founded an oil mill in 2004 in order to be able to make product refinement more independent (F7). The strategy of direct sales without on-farm processing of mixtures leads to a moderate return on investment, while if packaging and distribution of the mixture partners are on-farm, there is a higher potential of added value (F4). A forward vertical integration to secure the value chain through product refinement with its own sales in small containers was sometimes chosen as a strategy (F1, F4, F9). This highlights the challenges of mixtures and their profitability due to downstream value-added flows in the food supply chain.

### 3.1.6. Suitability for Farm-Specific Context

With the exception of F4, all interviewees farm according to organic principles and use measures that aim at aspects of closed nutrient cycles and minimized external inputs. Consequently, in these farms, PRP with respect to nutrient supply was clearly increased in favor of mixtures since ecological processes could substitute farming inputs, such as fertilizer and crop protection measures. Mixtures are grown on the farms visited at different locations with soil quality ranging from 11 (F3) to 70 (F5) soil points, according to the German soil classification scheme, where 100 soil points refer to the best soil. Particularly noteworthy is the suitability of mixtures for low-yielding soils with light sandy structure

(F3, F7). In addition, the location was mentioned as a reason for dissatisfaction with the previously implemented sole crop and its grain quality, so the cultivation of winter wheat-pea was adopted. However, winter wheat-pea is also cultivated on deep soils with up to 70 soil points (F5). Suboptimal conditions of shallow soils prone to silt and north-exposed slopes were also mentioned as good locations for mixtures (F1). Soil warming and thus nutrient mineralization are delayed in such cooler sites, making the provision of biologically fixed N during the flowering phase of wheat especially attractive. Other reasons for mixture advantages include reduced risk of nutrient leaching where annual precipitation is high (>850 mm (F1)).

### 3.2. Mixture Management Process: Challenges of Species Interactions, Mixture Attributes and Resources for Adaptation

The cultivation of species mixtures impacted the whole farming process, highlighting the need to look at multiple steps in the farming process but also downstream needs in the value chain beyond the farm. Here, we address mixture design and cultivation harvest and post-harvest procedures for food grain quality (Figure 3). It became evident that food grain mixtures cannot be treated all the same in the adoption process as they differ with respect to their attributes influencing the management process and their adoption.

#### 3.2.1. Rotation and Mixture Design

Crop rotation planning is considered an important building block for mixtures, especially from a plant health perspective, as the frequent occurrence of legumes in mixtures was viewed critically, potentially leading to legume fatigue and yield depression (F1, F4, F6). Self-compatibility of the brassica species false flax (*Camelina sativa*) was also raised as an open question (F7). In high-risk crop mixtures (e.g., with lentils), mechanical weed control is hardly possible and needs to be dealt with through crop rotation (F6, F9). If fertilization was not used, the focus was on planning the green manure crops before the mixed crop (F6, F9).

A broad range of traits and features were considered crucial for successfully combining crop species, varieties and their ratios in species in mixtures. These traits are a means to influence the species interactions that need to be managed in mixtures, especially competition, but also harvestability due to variation in maturity.

In the selection of varieties of organic canola-vetch mixtures, special attention is paid to the frost sensitivity of the vetch (F3). Oats as a supporting crop for lentils were considered too competitive (F4). To reduce competition with wheat, short pea cultivars with determined growth were selected (F5). The unavailability of uniformly maturing high-yielding lupine varieties for SM poses a problem (see above) (F8). Similarly, a French lentil variety used for culinary reasons is found to have very uneven maturity (F4), making the determination of harvest timing, as well as the required post-harvest management for wet-harvested batches, a challenge. In contrast, the non-simultaneous maturity between the oil flax and false flax is considered to be less problematic, as the low risk of shattering, as a characteristic of false flax, leads to high harvest elasticity (F2). Another advantage of false flax is its high plasticity in mixtures with respect to its plant height, with 0.6 m in oil flax to 1.20 m in oats (F2). Instead of vetch, buckwheat was used as a weed-suppressing mixture partner in winter canola (F3), but this mixture deprived plants of nutrients, making it vulnerable to canola pollen beetle (F3).

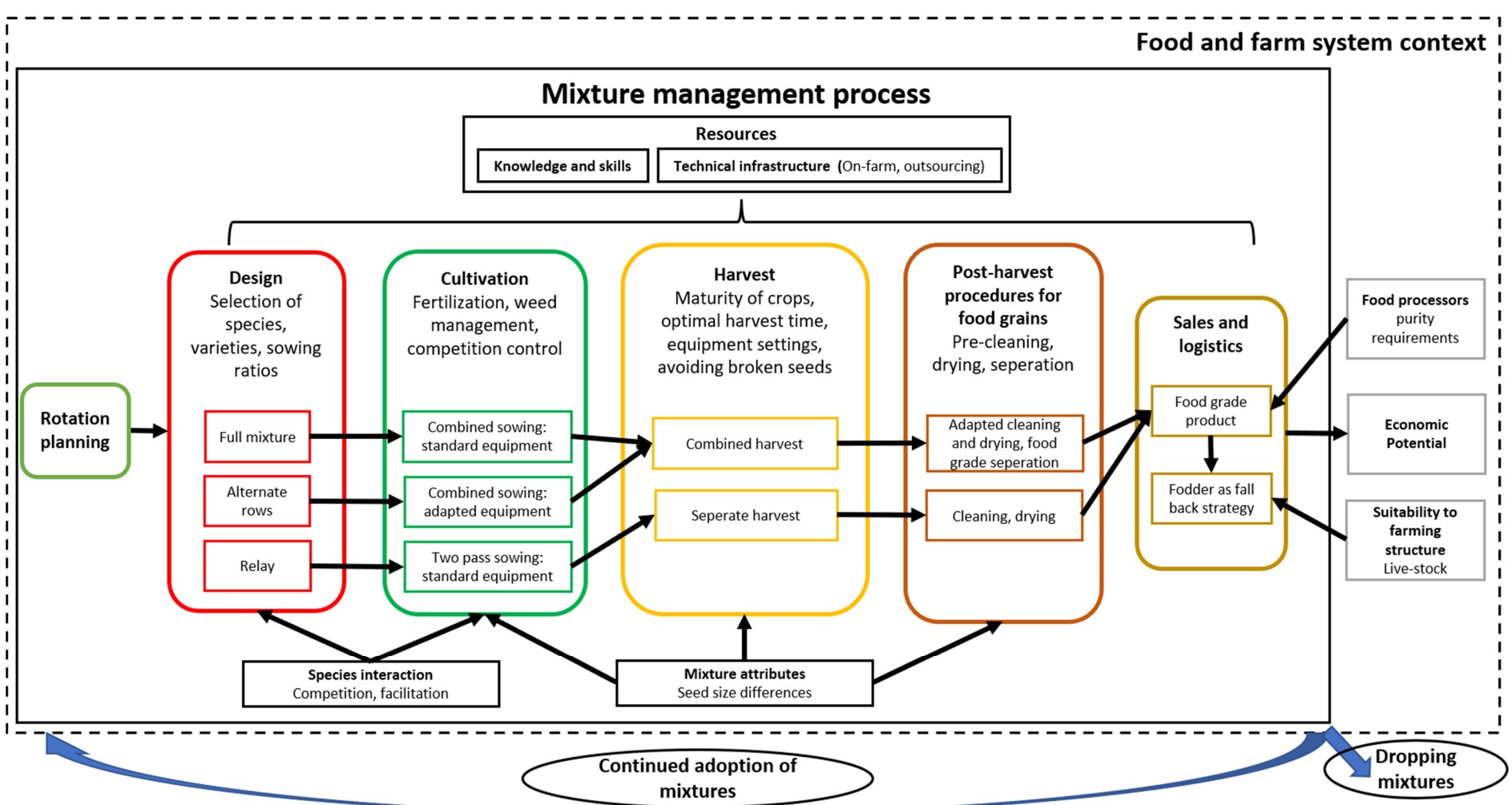

**Figure 3.** Social representation of the mixture management process in the context of the farming and food system. Colored frames show the steps in the management process and how choices in each step influences steps downstream in the process. Black frames show how species interactions, mixture attributes and resources influence mixture management. The social representation was interpretatively reconstructed from the interviews with farmers.

Optimizing the mixing ratio was crucial to managing species interactions. In mixtures, there is a trade-off between a dense cereal stand to reduce weed pressure (F2, F7) and the goal of high yields of valuable legume mixing partners (F6), high amounts of biologically fixed nitrogen (F8) and the lodging susceptibility of legumes (F1, F5). Knowledge about the site-specific sowing ratio that can only be achieved by experience was considered crucial (F1, F2, F8, F9), and test plots were established to determine this by F8, e.g., competition for growth factors considerably influenced optimization of the seed rate (F2). Thus, reducing the seed rate of oats in mixtures (10 kg/ha) has a positive effect on food grain quality as high hectoliter weights are achieved in mixtures at lower densities (F7).

### 3.2.2. Cultivation

Sowing Technique and Equipment

The requirements for the sowing equipment were considered unproblematic by F1 if grain size differences between the crops were small since they are drilled in one pass at the same depth, and the same cultivation machinery can be used for staggered or combined sowing operations, and the mixed cultures are in the same rows. A complicating factor when sowing mixtures are grain size and shape differences (Figure 1) and different sowing depth requirements of the crop species (F1, F4, F6, F7, F8). Nevertheless, almost all of the interviewees sow fully mixed with conventional technology. In the case of simultaneous sowing from a seedbox, the laborious handling of the mixing (F1, F2, F5, F6, F9) and the risk of segregation during sowing (F8) is an issue. In order to meet the requirements of the different sowing depths, staggered sowing, associated with more effort, was adopted (F7, F8, F9). Experience with false flax emergence after spreading it on the surface into the already sown crop was rated as either too strong or very weak (F9). This application method has been associated with uncontrollable field emergence, as noted before due to water-logging or dry soil (F7, F2).

Several farmers had already informed themselves about special seeding techniques for mixtures that are available for purchase and could name the advantages (F2, F5, F6, F7, F8). Nevertheless, no special seeding technology for the mixed culture was available on any of the farms as low-cost technical solutions for mixtures were preferred. Strategies, such as several successive sowing passes or manual application of fine-grained seeds, made it possible to forgo the need for special seeding equipment. The investment in technology was not made because it was perceived as "not profitable" or because the extra effort was possible due to small field sizes. Only F3 adapted an existing pneumatic seed drill with two seed tanks to enable species-separated row placement as this was not available for purchase. This allowed species-separated double rows of winter canola and a double row of freezing vetch. Compared to the combined within-row sowing, this optimizes the individual plant–plant spacing in the mixture. With this pattern, interspecific competition was enabled by mechanical competition control (see below).

Fertilization and Competition Control

Four farmers did not use fertilizer in the mixtures (F1, F7, F8, F9) except sulfur fertilization to enhance nodulation in lupin (F8). The remaining farmers used standard organic fertilizers, such as compost. A condition for successful canola cultivation is to meet the high nitrogen requirement in the fall that fosters strong plants, reducing the need for crop protection (F3). A further measure to regulate the growth of the mixed crop and manage interspecific competition was a one-time targeted fertilizer application to barley-lentils in spring (Figure 4). Using biogas slurry, weak barley was strengthened as a support crop in spring without causing damage to the lentils (F6). With regard to seedbed preparation for the mixtures, no special soil cultivation was performed.

The alternate row design of the canola-vetch mixture of F3 allowed for mechanical competition control. A standard bar roller was used to roll the entire surface of overly tall canola plants underneath. When vetch plants are too vigorous, the rotary hoe is used as a precise row regulator. This is used to selectively intervene in the growth of vetch as a

mixing partner. With the help of both special devices in crop management, the purpose of regulating interspecific competition in the winter canola-vetch SM was considered a good strategy (Figure 4).

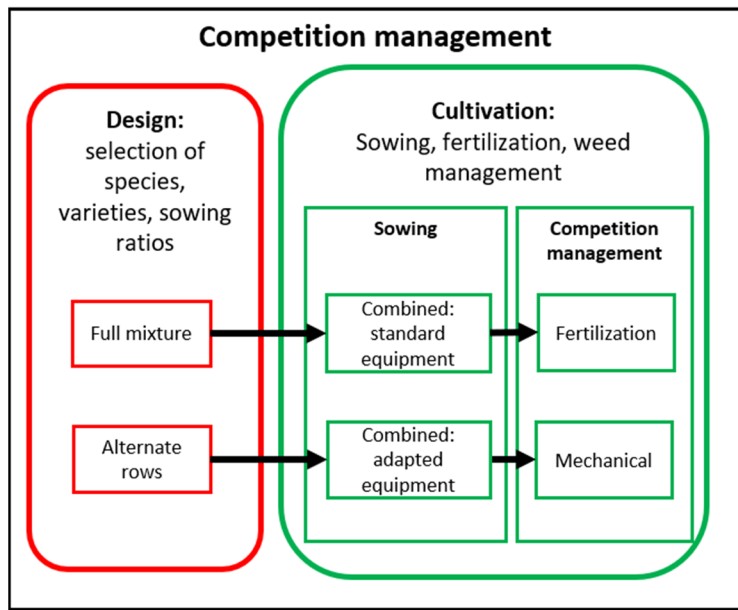

**Figure 4.** Social representation of the influence of spatial design of mixtures on competition management, interpretatively reconstructed from the interviews with farmers.

Weed Control

All mixtures for food grains are grown without herbicides on the farms surveyed. Therefore, only mechanical weed control is used (F1, F4), and the perceived attractiveness of mixtures was due to reduced weed pressure or simplified weed control. "That is also an advantage of the mixtures, that you don't have to do anything on the one hand, but, on the other hand, you can't do anything either" (F1). "For cultivation per se, the technical equipment is rather less critical than in the case of the sole crop. Because simply the weed control takes place much easier" (F8).

Winter mixtures are classified as relatively "clean" (F3, F5). In summer mixtures, such as lentil mixtures, there is a high weed pressure (F4). False flax is popular because of the rosette formation at the beginning of the plant growth as a "placeholder" instead of weeds (F2). Typical were statements about the "3-G" principle for SM: "gesät, geschaut, geerntet" ("sown, looked at, harvested", F6, F9) or "2-D" method: "drillen und dreschen" ("drill and thresh", F8). "The more mixture, the less harrow." (F1). Crop management was also considered "simple" (F6, F8) because SM required reduced mechanical weed control (F2, F7), thus saving time (F3). For example, F3 observed that late weed infestation, which makes technical harvesting impossible, would not occur in the canola-vetch mix if the soil was not moved after sowing. Nevertheless, where mechanical control is needed, especially in mixtures with grain legumes, such as peas (F5) and lentils (F1, F4, F6, F9), harrowing is demanding or a "tricky thing" (F9) and 10% damage to the crop must be accepted. Still, harrowing lentils was considered by one farmer (F4) as more relevant than the resulting losses.

3.2.3. Harvest Procedure and Equipment

Important challenges in the management of mixtures are the adjustment of the combine harvester and the timing that is usually based on the main crop (F5, F6, F7, F8, F9). In terms of adaptability for threshing close to the ground, a narrow combine was considered an advantage over wider ones for lentil mixtures (F4, F6). Challenges include damage to the equipment by stones, as well as the risk of stones in the threshed crop (F4). Grain loss at the

cutter-bar was one of the risks mentioned when harvesting mixtures with grain legumes (F1, F6), which can be somewhat alleviated by earlier harvest (F1, F8). Thus, frequently, very valuable crops were harvested earlier and wetter to avoid losses due to weather and shattering (F1, F5, F6, F8, F9). For example, wheat-pea at 18–19% (F5) and lentil-oats at 25% (F9) moisture. Higher contamination (F5) and increased moisture content (F1, F5) require drying and pre-cleaning in mixtures. Especially in dry conditions, reduced speed (F1, F4) can prevent grain loss (F6).

The greater the grain size differences, the more compromise has to be found in the combine settings (F2, F7, F8), which are time-consuming (F7). They influence the separability of the mixtures, especially if broken grains are the result (F8). The settings for mixtures with pea or lupin were described as relatively straightforward (F1, F5, F8), while lentil-barley is considered difficult (F6).

Experiences with contracted harvesters by the interviewed farmers were predominantly negative. "It's not just because of the technology, it's because of the mixture between technology that is too modern and that is self-adjusting, supposedly self-adjusting, and the drivers who just don't have the nerve" (F1). The special demands of mixtures with species susceptible to lodging could not be met and the modern electronic automatically adjusted technology can "hardly be outwitted" (F1). As pointed out above, wide machines with high speed are not suitable for mixtures from the farmers' point of view (F6).

### 3.2.4. Post-Harvest Procedures and Equipment

Due to the compromise during harvest with regard to the higher moisture content of the grains and the possible presence of weeds, pre-cleaning and drying were crucial in mixture management. The screen change between batches, monitoring of the product flow, control of the cleaning result and monitoring of the grain moisture in the drying process were explicitly mentioned as crucial for mixtures by all farmers. If these postharvest management procedures are not conducted in a timely manner, losses in food quality can occur. "And it has to work quickly, so not threshing today and tomorrow or the next day also think about how to dry. Threshing today, drying today, first priority." (F1). Food quality with heterogeneous dry matter in mixtures is achieved by pre-cleaning and multiple passes in the drying process (F6), as well as aeration and cooling (F2, F5). In terms of grain moisture, the requirements in the food sector are more demanding than those for feed mixtures (F5, F6, F7). Grain moisture of both mixture partners should be determined separately (F6) and, e.g., in the case of oat-linseed, drying of the two mixture partners even takes place separately (F7). Particularly in the case of oilseed crops for food use, immediate cleaning and drying are needed at >9% moisture (F7). The free fatty acid content is used as a measurable quality parameter for edible oils (F3, F7). Green weed seeds (F7) or undersown clover (F3) could have a negative influence on the taste of the oil.

A success factor for SM is the investment in special post-harvesting technology. Thus, pre-cleaning and on-farm drying facilities for food crops are considered to be necessary and were present at all farms (F1, F2, F3, F5, F6, F7, F8, F9) except F4. This equipment is relatively inexpensive and similar to that used for sole crops (F1). In the separation and processing of the false flax mixtures, only seed cleaning technology is used, whereby a combination of "sieve cleaning" with "pressure wind" or "aspiration" (F2, F7) without using weight and color separation (F7) is sufficient. An extension of the sieving equipment was also considered an important investment by F2, F7, F8 and F9. During the separation and cleaning of mixtures, there are several outlets from the "top sieve" or the "bottom sieve" containing valuable fractions. If these are not combined but discharged separately, it requires several conveying elements (F2). If low partial yields are expected from a mixing partner, special drying is used for small batches, e.g., a mobile truck drying system (F1, F3, F8). For pre-cleaned false flax batches, drying was set up for small batches in order to ensure food quality for as little as half a ton of harvest quantity (F7).

Over time, the complete equipment for ready-to-food products was established on four farms (F1, F2, F7, F9) or available through the lease of a second farm site (F5). F2 did

not invest in special cleaning technology, such as "vibrating screen [ . . . ] table separator or gravity separator", but costs were incurred due to increased logistic effort for processing by experienced colleagues. Optical sorting technology, in particular, was regarded as a "clever idea" but "capital-intensive" (F2). Failed marketing attempts occurred due to impurities or lack of a separation facility for the food grains (F2, F3). For the separation of lentil-oats, sieve cleaning, rotary cleaner (F1) and gravity separator were considered decisive (F9).

"So an example: Wheat with pea. You can get the pea out of the mixture by cleaning generously and leaving a few percent peas in the wheat. Then you have more or less pure peas. But then there are still five percentage points of peas in the wheat and then, of course, you can no longer sell the wheat as baking wheat." [ . . . ] "It doesn't work to clean it apart so well that the customer, i.e., the mill, which now wants to have the baking wheat, accepts the batches, because there are just . . . something is always in there" (F1). This phenomenon can be observed with broken peas and baking wheat (F1, F2), lentils and naked barley (F9), false flax with Rumex (F7), false flax with Chenopodium seeds (F4) and canola with *Galium* (F3). For better separability by grain size differences, lentils were combined with false flax, despite the better reduction of weed pressure by oats (F4).

F6 provides drying and separation services for a group of about 90 lentil producers in southern Germany. When batches could not be sufficiently cleaned, long distances were traveled by experienced colleagues to ensure successful reprocessing (F2). For contract drying, a minimum quantity of 20 tons is required to operate the drying technology (F4).

### 3.2.5. Food Grain Mixture Attributes

In this section, we zoom in on mixture attributes that influence mixture adoption and were reconstructed in the interpretative analysis of PRP and the mixture management process. Mixtures varied considerably with respect to their attributes, influencing the adoption process. The mixture attributes cannot be considered totally fixed but are socially constructed and vary over time and social context. For example, the crop value depends on the market conditions in the larger food system and the risk attributed to a mixture depends on the available management knowledge. In addition, attributes should be considered not as binary or discrete but as rather continuously distributed (Figure 5).

An essential mixture attribute was grain size differences in mixtures (Figure 1). Sufficient grain size differences enabled the successful separation of mixtures for food quality. Easiest to separate were lentil-false flax (F4), canola-vetch (F3) and barley-phacelia (F1). Furthermore, grain differences facilitate the adjustment of the rotary cleaner for lentil and barley (F6) and for lupine-barley mixtures (F8). With small grain differences, separation became more difficult. Another important mixture attribute was the degree of dependency of the crops on the mixed cropping system. Some crops have a nearly total dependency on the mixed cropping system and, therefore, constitute obligatory mixtures. This includes lodging-susceptible crops, such as lentils (F1, F4, F6, F9), long-stemmed peas and vetch (F1). In contrast, crops of optional mixture types can be grown as a sole crop as well. The crop value of the component crops has a major influence on the economic potential of the considered mixture in the farming system. Lentils are a high-value crop in contrast to wheat, and, therefore, this increases its economic potential even under increased efforts of cleaning and separation (F6).

The associated risk was a further key attribute of mixtures. The highest cultivation risk was attributed to lentil mixtures (F1, F4). In addition, lentil mixtures are counted among the expensive crops due to high seed costs (F1). The management challenges associated with this mixture were the increased risk of crop damage from weed control, the determination of harvest timing, combine harvester setting and the required pre-cleaning/drying of immaturely harvested mixtures. Another risk was the high fluctuation of the partial yields of lentils "because lentil is uncertain." (F4). The high perceived risk for some mixtures was, in part, due to the lack of available knowledge. Regarding yield reliability, the winter wheat-pea mixture was mentioned as a less risky or safer or even an easier mixed crop type (F1). This mixed crop was recommended for newcomers: "So my mixture, I rate it very safe

and very simple. In my opinion, everybody can start with it. You can't do much wrong with it" (F5). Higher yield reliability was also associated with this mixed crop type (F1, F5). "Whereas with winter pea and the winter cereals you always harvest something." (F5) This mixture was simpler to manage due to the similar cultivation technique for sole crops and the reduced weed control compared to pea sole crops (F2, F3, F5, F7, F8). Further, the uncomplicated adjustment of the combine (F1, F3, F5, F8) was mentioned. However, these assessments did not include the challenge of obtaining food-grade wheat from the mixture. Another less-risky mixed crop type was with false flax with regard to reduced weed control. False flax reached the optimal harvesting time earlier than its mixture partner (F2) with low risk of pod shattering, leading to high harvesting elasticity, so the cultivation of this mixed crop type is considered safe (F2, F7).

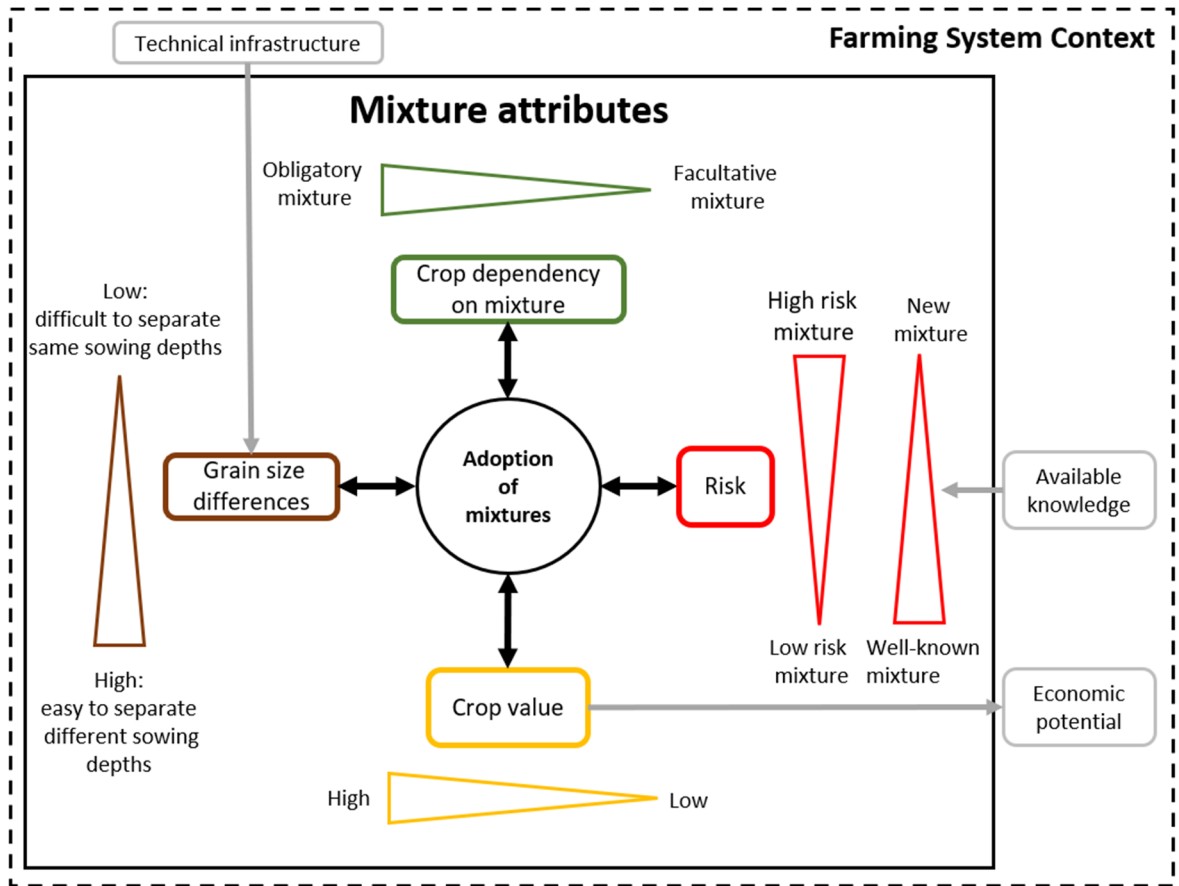

**Figure 5.** Social representation of mixture attributes and their influence on the adoption process in a farming system context, interpretatively reconstructed from the interviews with farmers.

Spatial arrangement is also important, as demonstrated by the canola-vetch mixture by F3 described above, which enabled specific control of competition but also required adapted sowing equipment.

### 3.2.6. Information Needs and Availability

A deficit of sources for specialized knowledge of the mixed culture was found in almost all interviews, except in F2. Sources of information on research conducted by a college or university on the increased promotion of biodiversity in lentil fields were mentioned (F6) or the latest French technical articles that mention cultivation variants with canola and vetch (F3). In order to find out the variety parameters for false flax, research results from an experimental farm are currently being used for the false flax variety comparison (F7). F2 learned about the advantages of mixtures as a cultivation strategy as part of

university education; however, there is the impression that the entire potential of possible combinations in mixtures has not yet been exhausted (F6). Consequently, experience-based knowledge was mentioned in all interviews as crucial but varied widely from 3 (F9) to 48 (F3) years (Table 1). During a test phase, it was examined to what extent the mixed culture was suitable for the operational farm conditions, and the benefits and effects of mixtures became visible because "own experience plays the biggest role" and "after two, three years [ ... ] you know how it works" (F1). Especially in the case of the mixed crop winter canola-vetch, it was necessary to collect one's own experiential knowledge because no contact person was known. There were no fields that could be visited and no consultants or publications that could be consulted in the development of mixtures (F3). "Yes, very weak. So that's where you actually hardly get information, even if the topic is so exciting, but I do not know any reliable sources of information. I don't know any sources of information there" (F5). Through years of experience, F6 perceived the development of the barley-linseed mixture as a "completely blind hinting" with "trial and error".

In all interviews, the network of colleagues experienced with mixtures was pointed to as a relevant source of information that had the potential to speed up adoption by avoiding "initial mistakes" and "If you have to work out everything yourself, you lose three or four years until you have experience of how it works." (F7). Topics that were explicitly mentioned in the exchange with experienced colleagues were variety and species selection (F4, F7), sowing ratios (F7), weed control (F9), adjusting the combine for fine-grained mixture partners (F7) and separation equipment (F2, F9), indicating a strong need for mixture-specific knowledge in cultivation. The information situation about lentil cultivation in SM is "very good" in southern Germany, as the farmers are networked (F9) and could be readily supplied from a producer association with information (F6). F4 observed that the literature described oats as a standard mixing partner for lentils, but they turned out to be too competitive. It was colleagues that suggested false flax as a suitable support crop. This hints toward context-sensitivity of mixture knowledge since the best fitting partners for lentils vary.

Organic advisors were considered to have expertise for simple/easy mixtures, such as winter wheat-pea, but not for "tricky" mixtures, such as lentil mixtures or oil flax-spring barley (F1). F2 consulted organic farming manuals, and another source of information was the website on organic agriculture. Farm managers from F3, F4 and F5 do not use official advice at present or when adopting mixtures on the farm. As further sources of information, several times articles in technical journals that attracted the farmer's interest in mixtures (F8) or gave hints to experienced colleagues (F2) were mentioned. The need for networking on the specific topic of mixtures was confirmed by the active participation in the mixture association in southern Germany that organized annual meetings with lectures and discussions and practical trials (F2, F7, F8). However, at the time of the interviews, the association was no longer active (F2, F8).

Uncertainty existed with respect to legume self-compatibility. Research and experiences of colleagues on this topic "simply do not exist" (F6). "One example, we are trying right now maybe lupin-oat mixtures ... I know a colleague who has tried it for a year, but I don't know at all how lupins and lentils behave in the crop rotation" (F6).

## 4. Discussion

This qualitative study of SM adoption enabled reconstruction of the main factors for food SM adoption in Germany. Ideally, the number of interviewed farmers would be higher, especially with respect to conventional farmers. However, no more conventional farmers could be identified meeting the criteria of at least three years of experience in food SM practice, as the use of food SM is even rarer in conventional farming. The aim was to identify experts in growing food SM that had at least three years of experience. Qualitative research standards, such as theoretical saturation [14], have largely been met for food SM adoption for organic farms. Including conventional farmers' perspectives could be highly relevant to mainstream agroecological approaches beyond organic farming [23]. This

would require another social science approach than expert interviews, e.g., agroecological design studies [24]. Future qualitative studies on food SM could include farmers from more European countries and actors in the food value chain. Whether a quantitative survey on food SM adoption meeting quantitative sampling standards is possible remains open since the adoption rate of food SM in Europe is unknown, to our knowledge. Despite some limitations, this study provides valuable new insights on food SM adoption relevant for Germany, Europe and beyond.

### 4.1. Conditions for Relative Mixture Advantages and Economic Challenges

Similar to general results from the innovations literature [13], perceived relative performance was the main reason for mixture adoption for the farmers interviewed. The broad range of aspects of PRP of mixtures indicates that SM performance was perceived multifunctionally. This is supported by empirical multifunctional SM evaluation [6]. Farmers that work under nutrient-limited conditions, especially in organic farming, focus on nutrient cycling and enhancing biological nitrogen fixation. That is why the focus was mainly on SM with nitrogen-providing legumes. Thus, the suitability of the farm structure is a precondition for mixtures to unfold their advantages relative to sole crops, as otherwise, nitrogen can be supplied from synthetic sources. Economic conditions, such as fertilizer prices, influence the attractivity of increased nitrogen efficiency enabled by mixtures. A shift towards higher energy costs for nitrogen fertilizers produced by highly energy-demanding chemical synthesis could increase the relative advantage of legume mixtures. Some legume species, such as soy and mung bean, are better adapted to droughts that are becoming more frequent in temperate regions due to climate change [25] and could contribute to improved resilience of food SM under increased environmental stress.

Relative mixture performances were described as very nuanced from the farmer's perspectives, and especially yields and food grain quality were considered critically. The additional steps necessary to meet the purity requirements of the downstream value chain strongly affect the economic potential. Thus, while a farm structure amenable for SM production is a necessary condition for mixture advantages *within* the farm, this is not sufficient for the whole value chain requirements that are determined *outside* of the farm. The results are similar to the findings by Rodriguez et al. [26], who found diversification practices to be of stronger advantage in ecological service dimensions but challenging in terms of economic viability with consequences for the need to support these practices by incentives and regulations [27]. While some advantages of SM, e.g., crop protection or efficient nitrogen use and provision, may in themselves provide economic advantages depending on the cost of these inputs, enhanced habitat for wildlife and other ecological services have no market value and currently do not enhance the economic potential. Aare et al. [28] also confirmed that farmers have to face economic challenges caused by their diversification practices.

High separation efforts can only be justified for very valuable crops such as lentils [8]. In contrast, for organic wheat, the revenue is much lower. Viguier et al. [8] calculated separation costs would consume the entire revenue of the wheat. This explains why mixtures are an established approach for lentils in Europe, while only a tiny niche for baking wheat in Europe [9]. However, addressing crop value as a key attribute of mixtures opens exciting possibilities. For example, additional high-value crops, such as spices, should be considered for species mixtures. Some farmers in Germany already grow caraway with barley in a relay cropping system, exploiting the high crop value of caraway with the mixture advantage of weed suppression (J.T., personal communication).

A fundamental economic challenge that is introduced by increased cropping system diversification is due to economies of scale [29] with respect to an increased number of crops in variable quantities requiring drying, separation, cleaning, storage and sales. In addition to higher costs per unit and reduced economies of scale, additional storage or transport facilities needed for small batches have adverse effects on profitability. A study investigating diversification practices in France also found that factors opposing

diffusion of diversification practices are economic scale effects, especially for logistics and grain processors [30]. However, economies of scale could be harnessed in diversified production systems by increasing the scale of diversified production systems, including centralization of post-harvest processing of mixtures as described for F6, who collects and cleans the produce for a multitude of partners. A key challenge seems to be the efficient combination of the ecological mixture advantages by diversification with economic viability that is conditioned by downstream needs in the food sector, which is highly dominated by economies of scale.

### 4.2. Handling Increased Cropping System Complexity to Harness Advantages

By reconstructing the mixture farming process, we showed that, compared to sole crops, SM adds complexity to the cropping system within a farm that affects the entire practical farming process from rotation planning, mixture design, cultivation, harvest and post-harvest, confirming the observations of Rodriguez et al. [26]. Several types of plant interactions, including competition, cooperation, compensation and complementarity—the 4C [31]—need to be managed in rotation planning, mixture design (choice of cultivar, species sowing ratios and patterns) and cultivation (fertilization, crop protection). Since ecological interactions, such as competition in mixtures, depending on local environmental factors, this adds another layer of complexity and is expressed in the challenge of highly variable partial yields in mixtures observed by farmers. In addition, mixture attributes, such as differences in seed size and maturity, as well as sensitivity in handling (e.g., broken grain), were reconstructed as emergent properties of SM that add further complexity to the cropping system. According to Rogers [13], the complexity of innovations is a main factor hampering their diffusion in society. The adoption of more complex innovations requires more knowledge, practical skills and adaptation with respect to the working process. These requirements can slow down the adoption process.

Increased cropping system complexity due to mixtures required specific knowledge by the farmers. This includes knowledge of suitable crop species and varieties, sowing ratios, weed control, fertilization, effects of mixtures in rotations on crop health, regulation of interspecific interactions and grain separation. Not only because of the many possible crop/variety combinations and context-sensitivity of this knowledge but also because extension and education largely neglect the topic, there seems to be a strong gap of information highlighted by all farmers, especially for the more sophisticated and new mixture combinations (vetch-canola, linseed-false flax/barley) and cleaning/separation issues, confirming earlier results [26,28]. To enhance mixture diffusion, there is a dire need to document and make mixture experiences available and integrate knowledge on mixtures into the education of farmers and advisors. Aare [28] suggested a farmer-to-farmer approach towards learning and knowledge for diversified cropping systems, recognizing farmers' potential as mixture experts.

Technological challenges were mostly overcome by the very creative farmers. While, in some cases, increased complexity reduced the required cultivation efforts, especially with respect to weed management, in order to enhance mixture performance, sowing and harvesting equipment, in particular, may need to be adapted. For sowing, mostly existing equipment for sole crops was used. To avoid multiple passes or unpredictable crop emergence due to the wrong sowing depth, as well as to optimize sowing patterns, adapted machinery for simultaneous sowing at different soil depths and/or alternating rows/strips is currently missing or too costly, again due to a lack of market size. The same applies to harvesting equipment capable of harvesting narrow strips or to adjust to mixtures of different seed sizes and sensitivities to avoid loss of or breaking seed.

Taking both together, the economic challenges for mixtures with their increased complexity and (technical) resources required to handle them explains the magnitude of challenges faced by species mixtures for food.

*4.3. Moving beyond Current Practice: Farm and Food System Level Innovations*

The increased complexity of a cropping system based on SM is the foundation for ecological interactions that provide relative advantages compared to sole crops. Therefore, the increased cropping system complexity and relative advantages are two sides of the same coin. Farmers have recognized that this broadens the option space for new mixture innovations on a crop system and harness mixture advantages better. Consequently, they have developed new mixtures that include high-value crops, such as spices, and spatial designs that decrease separation efforts, e.g., relay systems, or allow for better control of crop competition, e.g., alternate row design.

Lock-ins have often been diagnosed for diversification processes in our contemporary agricultural system [30,32]. If lock-ins are caused at a systems level, this requires a system perspective taking the multiple levels of agriculture and the food system into account. A food system turn in agroecology was suggested to facilitate such a complex perspective and to harness system level levers for diversification [33]. According to Gliessman [34], five levels need to be considered for a food systems turn: (1) increased resource efficiency, (2) substitution of conventional with alternative inputs, (3) redesign based on diversity and ecological processes, (4) establishing new links between agriculture and consumers via the value chain on regional levels and (5) a global systems perspective.

However, it is misleading to assume that these different levels of challenges need to be addressed in a stepwise fashion; rather, they need to be approached simultaneously [28]. For example, for food grain mixtures, only addressing the increased complexity in the cropping system (level 3) while disregarding the need to also address challenges in the downstream value chain for food grains, i.e., food processing and consumer perceptions (level 4), is likely to fail and might lead to the inappropriate conclusion that species mixtures are not economically viable beyond a tiny niche of special crops. From a food system perspective, it makes sense to address a higher level 4 (food processors) while the lower level is still not entirely optimized (managing cropping system complexity, level 3).

A prime example are wheat pea mixtures that are rarely used in practice [9] despite their manifold advantages, such as increased protein content in wheat and an additional yield of peas [6]. The usually discussed conundrum is the separation of baking wheat from mixtures. Bread baking usually relies on high-purity ingredients combined in precise recipes. Further, purity from allergens, such as gluten or legume proteins, may play a role in specialty markets. Specifically, half peas require a high separation effort due to their similarity in size and shape to wheat grains. Due to the relatively low revenue for wheat, such separation efforts can lead to prohibitively high costs. To reduce the separation effort, the key is to optimize the threshing process to avoid broken peas or to breed peas that do not break as easily. Taking the bakers into the equation can be an additional lever. Up to 10% peas in a baking mixture still result in good baking and taste properties [35]. Allowing for pea residuals of a defined proportion could decrease separation efforts and deliver a product with a defined composition without interfering with baking quality. Another potential added value would be a local, high-quality protein source. For example, in Germany, recently, the admixing of faba beans to improve the protein value of bread is being promoted with success, and faba bean flower is added to high-quality French baguettes to improve crust properties. Taking into account different quality standards for food can drastically reduce the required steps and effort in the separation process.

## 5. Conclusions

Species mixtures provide a range of advantages compared to sole crops from a farmers' perspective, but challenges remain with respect to the economic potential and food grain quality. Therefore, the adoption of food grain SM is highly dependent on very specific conditions, such as high-crop value, specific knowledge and technical equipment. To step out of the niche, the adoption of food grain SM needs to surpass these constraints. Here, we suggest three conclusions that could pave the way for food species mixtures in the future.

1. Increased complexity of cropping systems based on species mixtures is a basic property and needs to be recognized. This causes challenges in mixture management, such as required knowledge and technology, but is also the foundation of their advantages due to ecological interactions. The complexity of mixtures also enables the broadening of the option space for new cropping system design in terms of species combinations and spatiotemporal design. Recognizing this is a basic precondition to developing species mixtures for food beyond their current niche.

2. Following this prerequisite, education and learning on SM through various channels is one step forward for the adoption of SM for which knowledge already exists but is hardly accessible. Hitherto, farmers attempting to change towards more agroecological production principles are basically left alone. Specific knowledge provided by farmer-to-farmer learning, agricultural advisors and the practical literature is key levers to foster mixture adoption for food grains. New digital tools for mixture design could help to handle the increased option space of species mixtures. Knowledge of food SM could also be provided by public agencies or companies providing innovative SM products, such as seeds of cultivars adapted for SM.

3. To develop new species mixtures for food, a more systematic approach is needed that takes into account emergent properties of species mixtures and how these affect the practical farming process and crop profitability. Therefore, it is crucial to include farmers as well as the actors in the food system actors from the start. Promising food species mixture candidates, including crop species choice, spatiotemporal design and sowing ratios, would have to be designed based on agronomic and ecological principles, but always with the effects on the downstream processes in mind.

**Supplementary Materials:** The following supporting information can be downloaded at https://www.mdpi.com/article/10.3390/agriculture12050697/s1, Interview guide.

**Author Contributions:** Conceptualization, J.T.; methodology, J.T., T.S. and T.R.; formal analysis, J.T. and T.R.; interviews T.R.; resources, M.R.F.; writing—original draft preparation, J.T. and M.R.F.; writing—review and editing, J.T., M.R.F. and T.S.; visualization, T.R. and J.T. supervision, M.R.F.; project administration, M.R.F.; funding acquisition, M.R.F. All authors have read and agreed to the published version of the manuscript.

**Funding:** This research was a part of the project ReMIX "Redesigning European cropping systems-based on species MIXtures" funded by the EU's Horizon 2020 Research and Innovation Programme (Grant Agreement No. 727217).

**Institutional Review Board Statement:** Not applicable.

**Data Availability Statement:** Not applicable.

**Acknowledgments:** We thank Matthias von Ahn, Sven Heinrich and Odette Denise Weedon for their valuable support in the participatory process, as well as all ReMIX colleagues, especially Henrik-Hauggaard-Nielsen and Laurent Bedoussac for their incredible expertise with species mixtures.

**Conflicts of Interest:** The authors declare no conflict of interest. The funders had no role in the design of the study; in the collection, analyses, or interpretation of data; in the writing of the manuscript, or in the decision to publish the results.

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
