# Peer review of "Adoption of Food Species Mixtures from Farmers’ Perspectives in Germany: Managing Complexity and Harnessing Advantages"

_agriculture, doi:10.3390/agriculture12050697_

Round 1
Reviewer 1 Report
I have been finished to review the manuscript entitled "Adoption of food species mixtures from farmers’ perspectives in Germany: managing complexity and harnessing advantages". This is a new issue to mention SM as novelty. In my previous knowledge, monoculture without contamination (weed competition) is general practice for agronomy traits. There are so many concerning issues when mixed crops in single harvesting i.e. competition canopy, allelopathy, maturity, fertilizer supply, etc. By the way, the manuscript has already mentioned. I concerned on the real practice on SM, which is alternative way to develop the novel food ingredients in the near future. Also, I found some minor corrected issues as:
L84 Please enter the 2.1 subsection in the new line.
L121 (Flick 2002) Please check the style of reference.
L159-160 (.........) What?
L510-511 Error! Reference source not found. What?
L742 "(Saathoff in prep. 2022)" If it is unavailable to the access with doi, please delete it.
This is my opinion.
Author Response
Dear Reviewer,
thank you very much for your time, expertise, corrections and suggestions we integrated them all:
|
Comments reviewer 1 |
Reply/action taken |
|
L84 Please enter the 2.1 subsection in the new line. |
Done. |
|
L121 (Flick 2002) Please check the style of reference. |
Corrected. |
|
L159-160 (.........) What? |
Error corrected, reference to figure added (was missing). |
|
L510-511 Error! Reference source not found. What? |
This was due to not unlinking cross references before converting to pdf. Solved. |
|
L742 "(Saathoff in prep. 2022)" If it is unavailable to the access with doi, please delete it. |
Deleted. |
|
|
|
Reviewer 2 Report
This study is an interesting research. However, some points must be improved.
Lines 84-95: This study is the qualitative research rather than quantitative research. Moreover, in order to find out what are the factors influencing mixture adoption of farmer, why the statistical method were not applied in this study? Using only qualitative analysis is quite weak for the results. More explanations to support the methodology are required.
Lines 116-118: This study aim to reflect the farmers’ perspectives in Germany, but total of participants were too little. How only nine farmers can be effectively represented the farmers’ perspectives in Germany? Perhaps, it is inadmissible.
Lines 387-388: The organic farmers did not apply chemical fertilizers. What are the organic materials that they use to improve the plant growth?
Lines 589-599: Do the organic farms obtained any organic standard? If yes, what kind? How they archived the organic standard? The processes and supporting factors to meet the organic standard are very important, which should be explained. It will be useful for other farmers in other countries. If not, how to help them to achieve it?
Lines 609-617: In addition, the legume plants is less water consumption, drought tolerances and resistant to climate change. Please see in [Arunrat, N., Sereenonchai, S., Chaowiwat, W., Wang, C. 2022. Climate change impact on major crop yield and water footprint under CMIP6 climate projections in repeated drought and flood areas in Thailand. Science of the Total Environment. 807, 150741.] [Thomas, M., Robertson, J., Fukai, S., Peoples, M.B., 2004. The effect of timing and severity of water deficit on growth development, yield accumulation and nitrogen fixation of mung bean. Field Crops Res. 86, 67–68.]
Lines 669-670: “According to Rogers [13] complexity of innovations is a main factor hampering their diffusion in society.” This sentence is unclear, please elaborate it.
Lines 743-711: Based on only nine farmers, are these conclusions reliable? Do we need a lot more sampling population to draw the strong conclusion and represent the whole country?
Author Response
Dear Reviewer,
thank you very much for your time, expertise, corrections and suggestions we integrated them as follows:
|
Comments reviewer 2 |
Reply/action taken |
|
Lines 84-95: This study is the qualitative research rather than quantitative research. Moreover, in order to find out what are the factors influencing mixture adoption of farmer, why the statistical method were not applied in this study? Using only qualitative analysis is quite weak for the results. More explanations to support the methodology are required. |
We agree to the strength of statistical approaches. However, the goal of this manuscript is to explore the perspective and experiential knowledge of farmers. Statistical approaches would be an inappropriate methodological approach for this research goal, while qualitative approaches are established, valid and appropriate for this type of research question and established in applied agricultural science. This is already explained in the methods section. |
|
Lines 116-118: This study aim to reflect the farmers’ perspectives in Germany, but total of participants were too little. How only nine farmers can be effectively represented the farmers’ perspectives in Germany? Perhaps, it is inadmissible. |
We agree that the number of farmers is relatively low. However, in line with methodological approaches and theories in qualitative social sciences not the number of farmers interviewed but theoretical saturation, the depth of interviews and qualitative reconstruction are the main quality standards for qualitative standards, which we think are met in this manuscript quit well. We added a paragraph to the methods (theoretical saturation) and the discussion about the methodological limits of this study and how it could be overcome in the future. |
|
Lines 387-388: The organic farmers did not apply chemical fertilizers. What are the organic materials that they use to improve the plant growth? |
Some organic farmers (also the organic ones) did not use any (organic) fertilization in mixtures, as described in line 287. One farmer used sulphur (line 387), one biogas slurry (lines 392-393). To improve the information we added another sentence about organic fertilizers that was missing. |
|
Lines 589-599: Do the organic farms obtained any organic standard? If yes, what kind? How they archived the organic standard? The processes and supporting factors to meet the organic standard are very important, which should be explained. It will be useful for other farmers in other countries. If not, how to help them to achieve it? |
All achieved at least EU organic standards, we added that information in the methods section and a link to the EU regulation describing the standards. |
|
Lines 609-617: In addition, the legume plants is less water consumption, drought tolerances and resistant to climate change. Please see in [Arunrat, N., Sereenonchai, S., Chaowiwat, W., Wang, C. 2022. Climate change impact on major crop yield and water footprint under CMIP6 climate projections in repeated drought and flood areas in Thailand. Science of the Total Environment. 807, 150741.] [Thomas, M., Robertson, J., Fukai, S., Peoples, M.B., 2004. The effect of timing and severity of water deficit on growth development, yield accumulation and nitrogen fixation of mung bean. Field Crops Res. 86, 67–68.] |
Thanks for the valuable literature hints. We added Arunrat et al. to address the contribution of drought resistant legumes to mixtures. |
|
Lines 669-670: “According to Rogers [13] complexity of innovations is a main factor hampering their diffusion in society.” This sentence is unclear, please elaborate it. |
We elaborated this sentence and explained that higher complexity of innovations requires more knowledge, skills and adaptation of the farming process resulting in a slow down of the adoption process. |
|
Lines 743-711: Based on only nine farmers, are these conclusions reliable? Do we need a lot more sampling population to draw the strong conclusion and represent the whole country? |
We agree that even in a qualitative approach more farmers would be good to interview. However, theoretical saturation as a quality standard in qualitative social sciences was met and we added information on this in the methods section. In addition, the current number of farmers practically deploying mixtures for food beyond lentil-mixtures is really low due to the reasons identified (lack of knowledge, increased complexity). We added a paragraph in the discussion on the methodological limitations and how they could be overcome by future studies |
Reviewer 3 Report
Thank you for the opportunity to review the manuscript ” Adoption of food species mixtures from farmers’ perspectives in Germany: managing complexity and harnessing advantages”. I like the context of the introduction of food species mixture in common agriculture and not many publications have been addressing the psychosocial/sociological aspect of this issue. The authors were faced with the challenge of explaining the motives and outcomes of such a production. There are individual cases of their cultivation, but the author managed to present it in a systematic and simple way on a selected example of Germany. Generally, feedback from the producers is of great importance today for understanding the problems in agriculture and proposing some new solutions. I particularly like Figures and visualization that can be interesting not only as “scientific” literature but also as a practical guideline for farmers, practitioners and extension. Therefore the authors selected a good approach and presented some interesting results. Some general considerations of the paper require some further clarification in this regard:
- The authors mention that there are “On-farm experiments with species mixtures were conducted on four farms, establishing a stronger link to farmers and resulting in fruitful informal discussions …” are those farms among 9 that were selected for the survey. An additional question regarding SM does farmers make their own mixtures or SM can be purchased. Also 8 organic farms and one conventional seem to be a slightly unbalanced approach because it is a completely different cropping strategy.
- Main themes (lane 102) are set very broadly which is why I recommend reducing their number and being clearer in defining - some of the topics are overlapping like 4 and 5 and might be confusing because the focus could go in two directions to change technology of the SM or to introduce some new SM
- In addition the aims of the study comparison of SM and monoculture in not clear in this paper. Comparison of SM with monocropping is not possible because it requires growing both options for some period of time in paired trials. Also, it should be monocropping not monoculture. Monoculture implies a certain period of growing the same crop in the same place. So there is no data to support this and I'm not sure that this condition is met to be called monocultures. It is rather a comparison with sole crops in rotation.
- The manuscript is based on the semi-structured expert interview and their expertise. However only table 1 provides some key information on the farms. It would be good to provide at least some general – key questions addressed to farmers similar to studies in social sciences
- In table 1 authors use the term “specific cultivation technology –SCT” and this question in my opinion has not been properly defined. To understand the specific cultivation technology (in dependence to cropping systems) one should know what is the benchmark or at least common cultivation technology. It could be also sowing time or specific varieties is that covered with this interview?
- Line 260 “….. and high yield fluctuations” I would say high yield it just yield fluctuations over the years
- Moreover, it would be good to indicate which farmers use which SM in the same table not just in figure 1
- SM adoption phenomenon has not been sufficiently addressed from the point of climatic conditions/ climate
- Line 387-394 fertilization has not been properly explained here. “Four farmers did not use fertilizer in the mixtures… “ what about the others. How do those farmers manage their field in organic agriculture with a holistic approach, did they use some organic fertilizers
- Line 702-703 This statement contradicts the general position of the paper “of very low adoption rate of species mixtures in Germany“ which was also confirmed through the analysis of 9 farms that grow food SM.
- Line 765 The problem may also arise from the fact that the species that are grown in mixed cropping are selected for cultivation in pure stands and for higher yield, which is why we should work on the breeding of new genotypes that can be more suitable for SM type of cultivation.
- Please also look for some technical mistakes in the text Line 84, Line 159-160
Author Response
Dea Reviewer,
thank you for your time, expertise, corrections and further inputs. We integrated them as follows:
|
Comments reviewer 3 |
Reply/action taken |
|
The authors mention that there are “On-farm experiments with species mixtures were conducted on four farms, establishing a stronger link to farmers and resulting in fruitful informal discussions …” are those farms among 9 that were selected for the survey. An additional question regarding SM does farmers make their own mixtures or SM can be purchased. Also 8 organic farms and one conventional seem to be a slightly unbalanced approach because it is a completely different cropping strategy. |
Farmers from on-farm experiments and interviews were not the same since they did not fulfil the expert criterion of at least three years of experience with food mixtures. The farmers make their own mixtures as currently food mixtures can not be purchased as such. We added this information. We agree that the sampling is not very balanced. However, food mixtures so far are nearly exclusively used in organic farming to our knowledge and since the goal was to sample experienced farmers our sample reflects this situation. We added a new paragraph in the discussion on methodological limitations of the study and how that can be solved in future investigations. |
|
Main themes (lane 102) are set very broadly which is why I recommend reducing their number and being clearer in defining - some of the topics are overlapping like 4 and 5 and might be confusing because the focus could go in two directions to change technology of the SM or to introduce some new SM |
We agree that the main themes of the interview guide are broad. The full interview guide specifies much more precisely the addressed themes. To clarify we added the full interview guide as supplementary materials. |
|
In addition the aims of the study comparison of SM and monoculture in not clear in this paper. Comparison of SM with monocropping is not possible because it requires growing both options for some period of time in paired trials. Also, it should be monocropping not monoculture. Monoculture implies a certain period of growing the same crop in the same place. So there is no data to support this and I'm not sure that this condition is met to be called monocultures. It is rather a comparison with sole crops in rotation. |
Thanks for this clarification. We changed monocultures to sole crops. |
|
The manuscript is based on the semi-structured expert interview and their expertise. However only table 1 provides some key information on the farms. It would be good to provide at least some general – key questions addressed to farmers similar to studies in social sciences |
We added the interview guide as supplementary file. |
|
In table 1 authors use the term “specific cultivation technology –SCT” and this question in my opinion has not been properly defined. To understand the specific cultivation technology (in dependence to cropping systems) one should know what is the benchmark or at least common cultivation technology. It could be also sowing time or specific varieties is that covered with this interview? |
We agree our explanation was insufficient. We added a more precise definition to the caption of table 1: Specific cultivation technology refers to cultivation equipment specifically designed for species mixtures. |
|
Line 260 “….. and high yield fluctuations” I would say high yield it just yield fluctuations over the years |
We formulated now more precise “…high yield fluctuations observed over several seasons.“ |
|
Moreover, it would be good to indicate which farmers use which SM in the same table not just in figure 1 |
We added this to the table. |
|
SM adoption phenomenon has not been sufficiently addressed from the point of climatic conditions/ climate |
We agree that climate has not been a central focus in our study. However, in section 3.1.6 we addressed “suitability for farm-specific context“, including microclimatic conditions that are farm specific. |
|
Line 387-394 fertilization has not been properly explained here. “Four farmers did not use fertilizer in the mixtures… “ what about the others. How do those farmers manage their field in organic agriculture with a holistic approach, did they use some organic fertilizers |
We added some information about fertilization the farmers used for their sole crops. |
|
Line 702-703 This statement contradicts the general position of the paper “of very low adoption rate of species mixtures in Germany“ which was also confirmed through the analysis of 9 farms that grow food SM. |
Line 702-703 state: “they [farmers] have developed new mixtures that include high-value crops such as spices and spatial designs that decrease separation efforts, e.g. relay systems, or allow to better control crop competition, e.g. alternate row design.“
We see no contradiction of innovative approaches by few farmers with a currently slow adoption rate. |
|
Line 765 The problem may also arise from the fact that the species that are grown in mixed cropping are selected for cultivation in pure stands and for higher yield, which is why we should work on the breeding of new genotypes that can be more suitable for SM type of cultivation. |
We added breeding for mixtures to this paragraph. |
|
Please also look for some technical mistakes in the text Line 84, Line 159-160 |
Done. |
Reviewer 4 Report
The subject of the article is interesting, and it is linked to the objectives of the journal, however, there are some issues that have to be reconsidered.
For better visibility on databases, the authors are asked not to repeat among keywords the words/concepts included in the title of the article.
The literature review is well constructed and the references are appropriate.
The results could be interesting but they are not well discussed, and the conclusions sustain the results. It is advisable to create a distinct part for formulating general conclusions and recommendations for scholars, government, business etc.
Author Response
Dea Reviewer,
thank you for your time, expertise, corrections and further inputs. We integrated them as follows:
|
Comments reviewer 4 |
Reply/action taken |
|
For better visibility on databases, the authors are asked not to repeat among keywords the words/concepts included in the title of the article. |
Done |
|
The results could be interesting but they are not well discussed, and the conclusions sustain the results. It is advisable to create a distinct part for formulating general conclusions and recommendations for scholars, government, business etc. |
There already is a distinct section with conclusions. However, we missed the public and private sector and added both. |
Round 2
Reviewer 2 Report
Accept in present form.